# A Review of the Role of Pollen in COVID-19 Infection

**DOI:** 10.3390/ijerph20105805

**Published:** 2023-05-12

**Authors:** Nur Sabrina Idrose, Jingwen Zhang, Caroline J. Lodge, Bircan Erbas, Jo A. Douglass, Dinh S. Bui, Shyamali C. Dharmage

**Affiliations:** 1Allergy and Lung Health Unit, Melbourne School of Population and Global Health, The University of Melbourne, Carlton, Melbourne, VIC 3053, Australia; nidrose@student.unimelb.edu.au (N.S.I.);; 2Centre for Food and Allergy Research, Murdoch Children’s Research Institute, Parkville, Melbourne, VIC 3052, Australia; 3School of Psychology and Public Health, La Trobe University, Bundoora, Melbourne, VIC 3086, Australia; 4Department of Clinical Immunology and Allergy, Royal Melbourne Hospital, Parkville, Melbourne, VIC 3050, Australia; 5Department of Medicine, University of Melbourne, Melbourne, VIC 3052, Australia

**Keywords:** pollen, COVID-19, virus infection, SARS-CoV-2, severe acute respiratory syndrome coronavirus-2

## Abstract

There is current interest in the role of ambient pollen in the severe acute respiratory syndrome coronavirus-2 (SARS-CoV-2 or COVID-19) infection risk. The aim of this review is to summarise studies published up until January 2023 investigating the relationship between airborne pollen and the risk of COVID-19 infection. We found conflicting evidence, with some studies showing that pollen may increase the risk of COVID-19 infection by acting as a carrier, while others showed that pollen may reduce the risk by acting as an inhibiting factor. A few studies reported no evidence of an association between pollen and the risk of infection. A major limiting factor of this research is not being able to determine whether pollen contributed to the susceptibility to infection or just the expression of symptoms. Hence, more research is needed to better understand this highly complex relationship. Future investigations should consider individual and sociodemographic factors as potential effect modifiers when investigating these associations. This knowledge will help to identify targeted interventions.

## 1. Introduction

The severe acute respiratory syndrome coronavirus-2 (SARS-CoV-2 or COVID-19) pandemic has had a huge impact on the way in which we live. Due to most countries implementing strict COVID-19 restrictions, there has been a significant reduction in hospital presentations with asthma, general practitioner consultations, medication use, and severity of respiratory symptoms [1,2,3,4]. Wearing face masks [5,6,7], self-medication [3], less time spent outdoors [2,8], use of indoor air purifiers [9], and reduced ambient air pollution resulting from a lack of traffic (e.g., NO_2_, CO_2_ and particulate matter) [10] reduces the risk of not only COVID-19 infection, but also respiratory symptoms caused by environmental triggers such as pollen. For instance, studies have found that face masks are effective in reducing allergic rhinitis, especially nasal symptoms, in those with pollen allergies. Furthermore, people with asthma and/or hay fever were encouraged to self-medicate, if possible, to avoid contracting the virus at healthcare facilities so their symptoms were well-controlled. Nationwide lockdowns also resulted in people spending most of their time indoors, so there was minimal exposure to air pollution and pollen [11,12]. Whilst indoors, the increased usage of air purifiers also helped to filter out viruses as well as aeroallergens.

In the past few decades, changing climatic conditions coupled with poor environmental conditions, such as longer and more intense pollen seasons, air pollution, and heat waves, have substantively impacted public health. Our work has been instrumental in showing the detrimental associations between pollen and asthma, allergic diseases (including food allergy and eczema), and poorer lung function, even in individuals with no underlying allergic respiratory disease or pollen sensitisation [13,14,15,16,17,18]. When COVID-19 presented the world with a major public health challenge, scientists began to study its human-to-human transfer and whether meteorological conditions such as temperature and relative humidity played a part. Few studies have investigated the role of pollen in COVID-19 infection. Pollen components (e.g., pollen collected by honeybees and stored in hives) have exhibited promising activity for treatment against the virus [19], as some phenolic compounds, namely, flavonoids and non-flavonoids, have inhibitory properties. For instance, quercetin, a form of phenolic phytochemical found in pollen, has been shown to inhibit the entry of SARS-CoV into Vero E6 cells in vitro [20], while kaempferol and its glycoside can inhibit the 3a channel protein of the virus [21]. However, because the virus is spread via the dispersal of bioaerosols, pollen may also be a carrier of the virus and consequently increase the risk of infection. To date, several studies have investigated this relationship, but findings were mixed, with some reporting an increased risk of COVID-19 infection while others reported a protective or null association.

## 2. Aims of This Review and Search Strategy

Herein, we aimed to provide the current evidence on the role of airborne pollen in the risk of COVID-19 infection. To identify relevant papers, we used a systematic search strategy using the PubMed electronic database, with the combined search terms of “pollen” and “COVID”. Inclusion criteria were English language studies published up until 3 January 2023 that investigated the role of pollen in the risk of infection related to COVID-19. This included both epidemiological and experimental studies, and studies that investigated the COVID-19 entry receptor (angiotensin-converting enzyme 2, (ACE2)) as an outcome. We also checked the reference lists of related reviews to avoid missing any relevant papers.

In total, nine articles fulfilled the eligibility criteria (Figure 1). Most articles were excluded because they were not related to the aim of the review. For instance, they were either not related to the exposure (pollen) or outcome (COVID-19 infection), or they investigated the effectiveness of COVID-19 preventive measures on pollen-induced respiratory health outcomes (e.g., the use of face masks to reduce hay fever symptoms). All included studies are narratively synthesised below and summarised in Table 1. Where available, we extracted relevant data such as author (year), country, main findings, pollen levels, meteorological data, and study period. We also extracted the stringency index of the country of which the study was performed. The stringency index is a measure of the strictness of COVID-19 policies based on nine indicators, including lockdowns, travel restrictions, school and business closures, public gathering restrictions, and quarantine and vaccination policies [22]. We included this measure as part of the data extraction process because if the stringency index was high, then people were more likely to stay indoors and be less exposed to both pollen and COVID-19. Because the stringency index is a daily measure, we calculated the mean.

## 3. Role of Pollen in the Risk of COVID-19 Infection

Pollen is hypothesised to either increase, decrease, or have no effect on the risk of COVID-19 infection (Figure 2). It may reduce the risk by acting as an inhibiting factor towards viral infections [31], e.g., by downregulating ACE2 in the nasal epithelium [32] and subsequently protecting people especially those with asthma and allergic rhinitis against COVID-19 [26,33,34]. Conversely, pollen may increase the risk of COVID-19 transmission by acting as a carrier [25,35]. Experimental studies have shown that viruses can exist inside pollen grains or on its outer surface, allowing it to persist in the atmosphere for up to weeks and travel over long distances [33,36].

Damialis et al [23], who derived data from 31 countries across 5 continents, reported that pollen concentrations were associated with an increased risk of COVID-19 infection, with a lagged effect of up to 4 days. In the analysis, the authors accounted for the various lockdown types between countries (i.e., no lockdown, early lockdown, or later lockdown). They stated that pollen concentrations, together with humidity and temperature, accounted for, on average, 44% of the infection rate variability. Without a lockdown, infection rates increased by 4% for every 100 grains/m^3^ increase in pollen. Under similar pollen concentrations, lockdowns reduced the infection rates by half. Nonetheless, only pollen counts over 4 days (with a cumulative concentration of 1201 grains/m^3^) were studied, so the results could be due to chance, as the COVID-19 infections occurred in waves. In the same vein, Hoogeveen et al. [24] investigated whether environmental conditions (daily pollen concentrations, hay fever incidence, solar radiation, temperature, humidity, etc.) and mobility trends that were related to the seasonality of flu-like illnesses could also explain the seasonality of COVID-19 in the Netherlands [24]. This study had a mean stringency index of 57.2 and a mean pollen count of 69.2 grains/m^3^. Using backward stepwise linear regression, the authors found that a combined model of hay fever incidence, temperature, solar radiation, and mobility to indoor recreational locations accounted for 87.5% of the variance in the reproduction number of COVID-19 (R_t_). The authors did not include daily pollen concentrations into the combined model due to homoscedasticity issues, but pollen and hay fever were moderately to strongly correlated. Hence, similar results were expected if pollen data were to be used. Collectively, the findings from Damialis et al. [23] and Hoogeveen et al. [24] suggest that environmental factors such as pollen, temperature, and humidity could act jointly to explain the susceptibility towards COVID-19 infection. Besides the risk of infection, temperature and humidity have also been shown to influence pollen maturation, bio-aerosol formation, and flowering, so it is imperative to take into account these factors when assessing the association between pollen and risk of COVID-19 [37].

Pollen, which is a form of bioaerosol, may serve as a carrier for both bacteria and viruses [25] and/or reduce the innate immune responses, increasing susceptibility to infections [38]. To support the latter, studies have demonstrated that nasal birch pollen challenges can downregulate type 1 and 3 interferons in the nasal mucosa, and that the number of rhinovirus cases correlated with birch pollen concentrations. In the context of COVID-19, Shah et al. [26] showed that peak weed pollen counts (which had a maximum count of 200 grains/m^3^) preceded the peak of COVID-19 presentations (based on a 7-day moving average), lending further support to this argument. Alternatively, as most of COVID-19 infections were mild or asymptomatic, some studies may have detected more infections during the pollen season because of related symptoms such as runny nose or congestion. For instance, one study found that people were almost twice as likely to undergo COVID-19 testing on high-pollen days because their allergy symptoms were mistaken for those of COVID-19 [39]. Unless the whole population (including those without symptoms) were studied, it would be difficult to determine whether pollen really contributed to the susceptibility to infection or just the expression of symptoms.

On the other hand, pollen may have a protective role in the risk of COVID-19 infection. A study in the Netherlands described that pollen had inverse correlations with changes in flu-like incidence [31]. This study spanned 4 years from 2016 to 2020, so the study period included the COVID-19 pandemic. The mean stringency index was quite high (i.e., 60.75), suggesting that restrictions may have contributed to the reduced incidence. Aside from that, Jackson and colleagues, who evaluated different cohorts, have also implied that exposure to allergens, including pollen, could be protective against COVID-19 infection [30]. Because the study investigated different allergens, only findings related to pollen are outlined here. In the Urban Environment and Childhood Asthma (URECA) cohort, consisting of children, the authors showed that allergic sensitisation and increased levels of type-2 biomarkers such as IgE and IL-13 were able to downregulate ACE2 expression in airway cells, irrespective of asthma status [30]. This finding is consistent with the study by Gilles et al. [38], who found that pollen was able to increase protection against viruses, irrespective of allergy status. In another cohort, consisting of adults with mild asthma who were not on controller therapy, Jackson and colleagues reported that segmental allergen bronchoprovocation to ragweed pollen led to significantly lower ACE2 expression in the lower airway epithelium [30].

However, it is possible that factors other than ACE2 expression regulated the response in these individuals. As such, a Japanese study found no significant change in ACE2 expression in patients with seasonal allergic rhinitis induced by Japanese cedar pollen [29]. Therefore, teasing out other potential factors may also offer crucial insights into the COVID-19 disease pathogenesis. Besides ACE2, toll-like receptor-4 (TLR4) may also be operative, because it can bind to the virus’ spike protein with greater affinity than ACE2 [40].

Some studies have reported null associations between pollen and the risk of COVID-19 infection. A study in Spain replicated the findings of Damialis et al. [23], but this time, over a year (March 2020–February 2021), and observed no evidence of a relationship between daily ambient pollen concentrations and daily COVID-19 cases [27]. However, it was unclear whether appropriate modelling was conducted, as only correlation coefficients were reported in the paper. This study had a mean stringency index of 67.08, but did not report any pollen concentrations, so the lack of association could be due to the strict COVID-19 policies and/or lower pollen levels. In addition, the COVID-19 immunisation campaign in Spain was launched in late December 2020, so the null association could be because more people were vaccinated at the time of the study. Furthermore, Dunker et al. [28] indicated that pollen could not be a virus carrier as they observed no virus particles in ambient pollen during the COVID-19 pandemic in the city of Leipzig, Germany, but the infection rates in this country were low.

## 4. Conclusions

In conclusion, there is still limited evidence on the relationship between pollen and the risk of COVID-19 infection. It is difficult to determine a causal relationship because longitudinal, prospective cohort studies investigating these associations are lacking, with almost all existing studies being cross-sectional or experimental. A major limiting factor of this research is not being able to determine whether pollen contributed to the susceptibility to infection or just the expression of symptoms, unless entire populations, including people with no symptoms, are studied. Additionally, mechanistic immunological studies provide contradictory data on immune modulation. It is also important to investigate factors other than ACE2 in regulating immune protection against COVID-19. Furthermore, future research should consider effect modification by individual and sociodemographic factors when investigating the relationship between pollen and COVID-19 infection. For instance, obesity has been shown to interact with low lung function to increase the risk of infection [41], and the COVID-19 infection rates have been shown to be disproportionately higher among socially disadvantaged and/or some ethnic populations [42,43]. Hence, more research is needed in this area, as the pollen–virus relationship and interactions with other factors are still poorly understood and highly complex.

## Figures and Tables

**Figure 1 ijerph-20-05805-f001:**
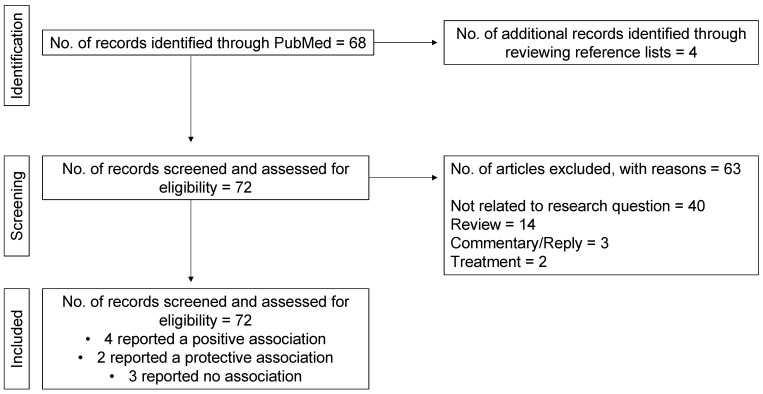
Flow diagram of the screening process.

**Figure 2 ijerph-20-05805-f002:**
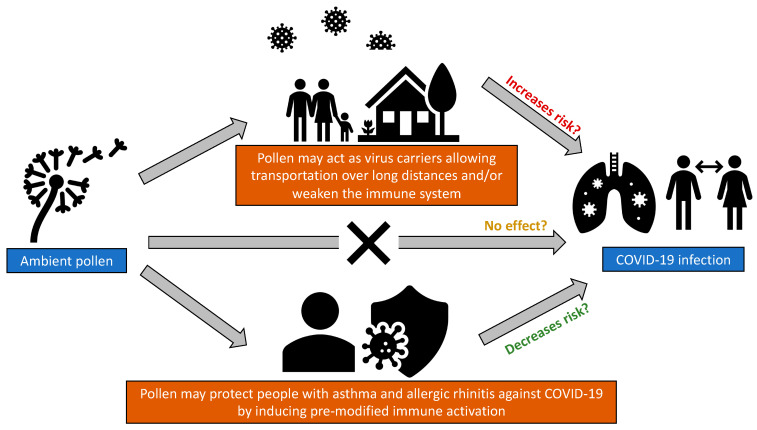
Hypotheses on how ambient pollen could impact COVID-19 infection.

**Table 1 ijerph-20-05805-t001:** Characteristics and main findings of included studies.

Author (Publication Year)	Country/Countries	Main Findings	Pollen Levels	Meteorological Factors	Study Period	Stringency Index	Statistical Methods
Positive association—pollen is associated with an increased risk of infection							
Damialis et al. (2021) [23]	31 countries across 5 continents (South Korea, Sweden, Ukraine, Australia, Belgium, France, Germany, Hungary, Italy, Latvia, Netherlands, Russia, South Africa, Spain, Switzerland, UK, USA, Argentina, Canada, Croatia, Czech Republic, Denmark, Finland, Greece, Lithuania, Poland, Portugal, Serbia, Slovakia, Slovenia, Turkey)	Pollen concentrations were associated with an increased risk of COVID-19 infection, with a lagged effect of up to 4 days. Pollen concentrations accounted for, on average, 44% of the infection rate variability.	Cumulative pollen concentration of 1201 grains/m^3^ over 4 days	Temperature and humidity were found to be significantly positively correlated with infection rates, indicating that they may act in synergy with pollen.	4 days (10 to 14 March 2020)	Mean values: South Korea (55.56), Sweden (26.11) Ukraine (34.44), Australia (19.44), Belgium (26.48), France (45.09), Germany (32.87), Hungary (43.15), Italy (81.85), Latvia (26.29), Netherlands (38.89), Russia (27.13), South Africa (13.89), Spain (50.09), Switzerland (28.33), UK (11.85), USA (27.59), Argentina (22.22), Canada (17.04), Croatia (27.04), Czech Republic (48.15), Denmark (47.22), Finland (30.00), Greece (40.55), Lithuania (no data until 20 March 2020 at 81.48), Poland (36.48), Portugal (40.10), Serbia (22.78), Slovakia (46.85), Slovenia (25.74), Turkey (23.15, no data on 10 March 2020).	Correlation analysis
Hoogeveen et al. (2022) [24]	Netherlands	A combined model of hay fever incidence, temperature, solar radiation, and mobility to indoor recreational locations accounted for 87.5% of the variance in the reproduction number of COVID-19 (R_t_). The authors did not include daily pollen concentrations into this combined model due to homoscedasticity issues, but pollen and hay fever were moderately strongly correlated.	Mean of 69.2 grains/m^3^	Temperature was only associated with R_t_ if mobility trends and pollen dispersion/maturation were taken into account. Humidity, in general, was associated with R_t_ and seasonal allergens.	7 months (17 February 2020 to 21 September 2020)	Mean (SD): 57.22 (18.12)	Backward stepwise multiple linear regression
Dbouk et al. (2021) [25]	Middle East	Pollen can increase the transmission rate by acting as a carrier.	Not relevant	The wind speed, temperature, and humidity were set to be 4 km/h, 22 °C, and 50%, respectively.	Not relevant	Not relevant	Computational multiphysics, multiscale modeling, and simulations
Shah et al. (2021) [26]	USA	Peak weed pollen count preceded the peak of COVID-19 presentations.	Graph showed a range of 0–200 grains/m^3^.	Meteorological data were not available.	4 months (18 July 2020 to 18 November 2020)	Mean (SD): 65.48 (2.66)	Nonlinear least squares regression
No association—pollen is not associated with risk of infection							
Moral de Gregorio et al. (2021) [27]	Spain	No evidence of a relationship between daily ambient pollen concentrations and daily COVID-19 cases.	Not stated	Meteorological data were not available.	1 year (1 March 2020 to 28 February 2021)	Mean (SD): 67.08 (13.01)	Spearman correlation
Dunker et al. (2021) [28]	Germany	No virus particles were found in ambient pollen during the pandemic.	Graph showed a range of 0–1800 grains/m^3^.	Meteorological data were not available.	4 months (1 Jan 2020 to 21 May 2020)	Mean (SD): 47.60 (29.11)	Purified pollen was tested for the presence of the virus using RT-PCR and virus-induced cytopathic effects (CPE) on Vero cells
Takabayashi et al. (2022) [29]	Japan	Mean levels of ACE2 expression in patients with seasonal allergic rhinitis induced by Japanese cedar pollen (JCP) did not significantly increase during the JCP season.	Not relevant	Meteorological data were not available.	Not relevant	Not relevant	Kruskal–Wallis analysis of variance with Dunnett post hoc testing and the Mann–Whitney U-test
Protective association—pollen is associated with a reduced risk of infection							
Jackson et al. (2020) [30]	USA	In the URECA cohort, consisting of children, the authors showed that allergic sensitisation and increased levels of type-2 biomarkers such as IgE and IL-13 can downregulate ACE2 expression (i.e., a COVID-19 entry receptor) in airway cells, irrespective of asthma. In another cohort, consisting of adults with mild asthma who were not on controller therapy, segmental allergen bronchoprovocation to ragweed pollen led to significantly lower ACE2 expression in the lower airway epithelium.	Not relevant	Meteorological data were not available	Not relevant	Not relevant	Weighted linear mixed effects model
Hoogeveen et al. (2021) [31]	Netherlands	Pollen had inverse correlations with changes in flu-like incidence, which included the COVID-19 pandemic period.	Mean of 349 grains/m^3^	Temperature was associated with pollen levels, but not flu-like incidence. Humidity and solar radiation were both associated with pollen levels and flu-like incidence.	4 years (4 Jan 2016 to 3 May 2020) COVID data were available from 27 Feb 2020 to 3 May 2020.	Mean (SD): 60.75 (27.51)	Linear regression

Not relevant because it was an experimental study.

## Data Availability

Not applicable.

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
