# Peer review of "A Review of the Role of Pollen in COVID-19 Infection"

_ijerph, 2023, doi:10.3390/ijerph20105805_

Round 1

Reviewer 1 Report

The paper investigates the role of ambient pollen in SARS-CoV-2 infection risk through a literature review.

Despite the topic cannot be developed in depth because of its complexity and limited evidence, the thesis that support a protective role of pollen are interesting. 

Discussion could be improved in its form. It could useful to make it a little more schematic.

At line 46 correct "sensitization". 

Author Response

We thank the reviewer for their time in reviewing this paper. Please see attachment for our response. 

Reviewer 2 Report

Overall, the paper is well written and interesting to read, however I see the following issues that should be resolved before publishing this paper:

Table 1. Characteristics and main findings of included studies, authors should analyse and add more details of meteorological data such as T, RH and wind speed), Theses factors could be affected on pollen and increased risk of infection.

Author Response

(The authors gave the same response as above.)

Reviewer 3 Report

This manuscript entitled “A review of the role of pollen in COVID-19 infection ” mainly describes the relationship between the measured pollen levels and covid-19 in the literature. The valuable data are helpful and essential in understanding of effects of pollens for the spreading of the covid-19. However, more studies are needed to explain the complex relationship between pollen and covid-19. Therefore, the contribution of this review study, in which only 9 studies were examined, to the literature was found to be limited.

Author Response

(The authors gave the same response as above.)

Reviewer 4 Report

This paper reviews the studies that investigated the relationship between airborne pollen and the risk of COVID-19 infection. This paper contains some problems, which should be fixed before publication.

1.  The abstract should include the conclusion of this review.

2.  The keywords are suggested to be more specific.

3.  Page 3 is blank.

4.  Fig. 1 should be upgraded, the flow chart does not show the steps or the relations very clearly.

5.  Pollen affects the COVID infection indirectly, but how to assess the relationship between pollen and COVID infection should be mentioned in this paper. The physical models of the three modes are suggested to be explained.

6.  The relationship between pollen and other factors (such as stringency index ) should be explained.

Author Response

(The authors gave the same response as above.)

Round 2

Reviewer 3 Report

Although there are different findings and a limited number of studies in the literature, the subject is interesting and gained more scientific soundness after revising the article.